# Ionospheric Nighttime Enhancements at Low Latitudes Challenge Performance of the Global Ionospheric Maps

Yuyan Yang [1,2,3], Libo Liu [1,2,3,*], Xiukuan Zhao [1,3], Haiyong Xie [1,3], Yiding Chen [1,2,4], Huijun Le [1,2,3], Ruilong Zhang [1,2,3], M. Arslan Tariq [1,2,3,5] and Wenbo Li [1,2,3]

1   Key Laboratory of Earth and Planetary Physics, Institute of Geology and Geophysics, Chinese Academy of Sciences, Beijing 100029, China; yangyuyan18@mail.iggcas.ac.cn (Y.Y.); zxk@mail.iggcas.ac.cn (X.Z.); xiehy@mail.iggcas.ac.cn (H.X.); chenyd@mail.iggcas.ac.cn (Y.C.); lehj@mail.iggcas.ac.cn (H.L.); zhangruilong@mail.iggcas.ac.cn (R.Z.); arslan.tariq@ncp.edu.pk (M.A.T.); llwwb@mail.iggcas.ac.cn (W.L.)
2   College of Earth and Planetary Sciences, University of Chinese Academy of Sciences, Beijing 100049, China
3   Heilongjiang Mohe Observatory of Geophysics, Institute of Geology and Geophysics, Chinese Academy of Sciences, Beijing 100029, China
4   Beijing National Observatory of Space Environment, Institute of Geology and Geophysics, Chinese Academy of Sciences, Beijing 100029, China
5   Centre for Earthquake Studies, National Centre for Physics, Islamabad 44000, Pakistan
*   Correspondence: liul@mail.iggcas.ac.cn

**Abstract:** In this study, two ionospheric nighttime enhancement (INE) events at low latitudes are selected to investigate their spatial features through the observations from Global Navigation Satellite System (GNSS) receivers and ionosondes. For the first time, we present the detailed spatial pictures of premidnight and postmidnight INEs under geomagnetically quiet conditions. The two INE events have the maximum extents of about $11° \times 34°$ and $17° \times 25°$ (longitude $\times$ latitude), respectively. Dramatic latitudinal and longitudinal features are revealed in the two INEs. We perform a comparison between the products of Global Ionospheric Maps (GIMs) and total electron content (TEC) measurement from GNSS receivers. However, GIMs fail to capture the TEC distribution during INEs owing to their limited spatial and temporal resolution. Considering the extent of INEs from the observations, the spherical harmonic (SH) expansion adopted by the GIM models needs to upgrade the degree and order to 36. The pixel-based methods developed from two GIM models are required to reduce their grid size for higher spatial resolution. The recommended time interval is shorter than 30 min. Among seven GIMs, CODG and JPLG maps generally have the best performance in reproducing the latitudinal structure of the ionosphere.

**Keywords:** ionosphere; ionospheric nighttime enhancement (INE); Beidou geostationary orbit (GEO); total electron content (TEC); Global Ionospheric Map (GIM); ionosonde

## 1. Introduction

As one of the most popular products in the space geodesy, Global Ionospheric Maps (GIMs) are produced by the Ionosphere Working Group (IWG) of the International GNSS Service (IGS) since 1 June 1998 [1–3]. The GIMs provide global Total Electron Content (TEC) from hundreds of global distributed permanent Global Navigation Satellite System (GNSS) receivers. TEC is one of the essential parameters of the ionosphere and can be estimated from GNSS dual-frequency measurements in terms of the dispersion properties of the ionosphere. The global snapshots of TEC from GIMs have been used in the science and technology fields. They supply global information for monitoring spatial and temporal behavior of the ionosphere [4–6] and mitigating ionospheric effects in precise positioning techniques [7,8]. The errors of GIMs might bring about less reliable understanding in scientific researches and further badly affect the GNSS applications. There always is a strong need to improve the performance of GIM models.

In nature, the electron density of the ionosphere experiences complex variations with solar radiation, chemical loss, dynamic transport, and so on [9]. Besides, the distribution of GNSS receivers is inhomogeneous, as oceanic regions and southern hemispheric continents have relatively sparse receivers. Thus, it is difficult to accurately represent the ionospheric behavior by GIMs. Recent studies aimed to assess GIMs by several mathematical methods, like VTEC-altimeter and dSTEC-GPS assessments [10–13]. They mainly focused on the accuracy of data values of GIMs on a long time scale. However, the performance of GIMs has been scarcely investigated from the point of view of the ionosphere. Jee et al. [10] compared GIM with TOPEX/Jason TEC data to assess the performance of GIM over the global ocean. They found that GIM was not accurate enough to represent ionospheric structures, such as equatorial anomaly, the wave-like longitudinal structure, and the Weddell Sea Anomaly. GIM models have continually been improved since the increasing number of TEC data being used and updated modeling techniques in the past decades. Therefore, the performance of recent GIMs in representing the ionosphere with high temporal and spatial variations is still pending.

The electron density in the ionosphere shows a strong diurnal variation [9]. In the absence of solar radiation, electron density in the nighttime ionosphere is expected to decay steadily. In some ionospheric models, nighttime behaviors of the ionosphere are often ignored. For instance, the nighttime TEC in the Klobuchar model is described as a constant bias of 5 ns [14]. Interestingly, many researchers have shown that the electron density in the nighttime ionosphere may increase frequently [15–26]. The phenomenon is commonly known as Ionospheric Nighttime Enhancement (INE). The amplitudes of INEs can reach up to several TECU (TEC Unit, 1 TECU = $10^{16}$ electrons/m$^2$), which could cause significant ionospheric delays. Therefore, it is necessary to represent the INEs in GIMs correctly.

The INEs have complicated behaviors with the local time, location, season, solar, and geomagnetic activity levels. Observations of NmF2 (maximum electron density of F2 layer) show that the occurrences of INEs have two peaks at premidnight and postmidnight, respectively, regardless of the season and solar activity level [27]. The premidnight and postmidnight INE events have different morphology and driving processes [9]. The spatial features of INEs have been investigated in many studies [28–33]. For example, Farelo et al. [27] illustrated the global morphology of INEs in NmF2 in the latitude range 15–60°N. Luan et al. [30] reported the significant enhancements along the latitude in different sectors from the COSMIC radio occultation observations. Chen et al. [31] showed spatial variations of the INEs in NmF2 on a global scale, varying with the season at solar minimum. Owing to sparse ionospheric observations, a statistical method is commonly employed by previous investigations. A detailed spatial picture of an INE event is still lacking, especially the longitudinal features. Thus, the detailed longitudinal and latitudinal evolutions and fine spatial extents of INE events still need further investigation. Nowadays, the dense GNSS observations provide us an excellent opportunity to give possible answers.

The aim of this paper is to investigate the detailed spatial features of specific INE events and evaluate the performance of GIMs on these INE events. In this work, we select two INE events under geomagnetically quiet conditions, with a strong premidnight and postmidnight INE at low latitudes in the East Asian-Australian sector, respectively. First, we analyze the horizontal spatial pictures of the INEs through measured values. Then, the characters of INEs in GIMs TEC are investigated and compared with the observed TEC.

## 2. Data and Methods

We select two prominent INE for different case studies designated the event on 2 December 2020 as Case A and 15 November 2020 as Case B, respectively. GIM data are analyzed with two different types of observations to depict the features of INEs.

Currently, there are seven Ionosphere Associate Analysis Centers (IAACs) of the IWG for the GIM products [34]. These are the Center for Orbit Determination in Europe (CODE) [3], Jet Propulsion Laboratory (JPL) [35], European Space Agency/European Space Operation Center (ESA/ESOC), Polytechnical University of Catalonia (UPC) [36],

the Energy, Mines and Resources/Natural Resource Canada (EMR/NRCan), the Chinese Academy of Sciences (CAS) [34], and Wuhan University (WHU). Individual IAACs generated the rapid, final, and predicted products with a time spacing of 15 min to 2 h. The TEC values are given at fixed grid points with a spatial resolution of 5° × 2.5° in geographic longitude and latitude, respectively. Different centers develop different mapping and interpolation techniques. JPL employs the three-shell model and Kalman filter approach [37] while UPC utilizes a voxel-defined two-layer tomographic model with splines and kriging interpolation [2,38]. CODE [3], ESA [39], and EMR adopt the spherical harmonic (SH) expansion and modified standard single-layer model (MSLM). Later, WHU applies an inequality-constrained least squares method with SH functions [40] and MSLM, and CAS uses the SH expansion plus Generalized Trigonometric Series (GTS) function (SHPTS) to generate their global TEC maps [41]. The GIMs from seven IAACs have been assessed by several methods, and the consistency among the seven GIM products is quite good [11–13]. The performances of the seven GIM products are analyzed in this work.

Two types of observations are utilized in this work: F layer observation from ionosondes and the ground GNSS TEC data. Figure 1 shows the locations of the ionosondes and GNSS receivers used in this study. Ionosondes are operated at Sanya (109.6°E, 18.3°N), Guilin (110.3°E, 25.3°N), Wuhan (114.4°E, 30.5°N), Beijing (116.2°E, 40.3°N), and Mohe (122.5°E, 52.0°N), respectively. The coordinates in this paper are given in geographic coordinate system unless otherwise stated. The ionogram traces are manually scaled to provide the height profiles of electron density, foF2 and hmF2 parameters at a time interval of 15 min. The electron density height profiles Ne(h) are derived with the built-in SAO-explorer [42]. foF2 and hmF2 are the critical frequency and the peak height of F2 layer, separately. The critical frequency (in MHz) is proportional to peak electron density of F2 layer (NmF2 in $cm^{-3}$, NmF2 = $1.24 \times 10^4 (foF2)^2$). There is occasional missing of foF2 and hmF2 observations due to the severe spread-F.

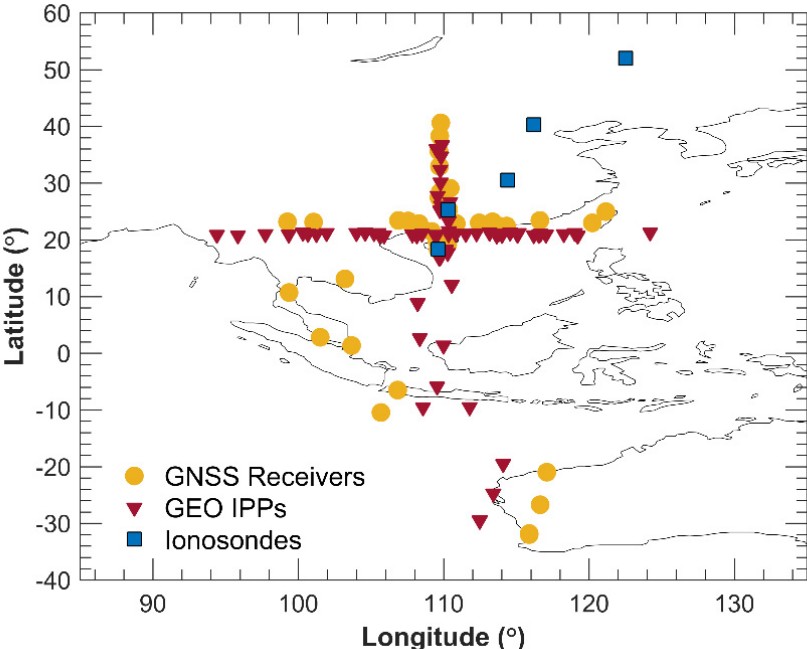

**Figure 1.** Locations of the ionosondes (blue squares) and GNSS receivers (yellow dots) used in this study. Red triangles mark the ionospheric piercing points (IPPs) of BeiDou GEO satellites paired with GNSS receivers.

We utilize the TEC data at a time resolution of 30 s from the BeiDou Ionospheric Observation Network (BION) from the Institute of Geology and Geophysics, Chinese Academy of Sciences (CAS), Beijing, China [43], the Chinese Meridian Project [44], and the IWG of the IGS [45]. These GNSS receivers are symbolized by yellow dots, as shown in

Figure 1. Slant TEC (STEC) is the integral of electron density along the GNSS signal path between the satellite and the receiver. Vertical TEC (VTEC) is the projection of STEC on the zenith direction under a thin-layer assumption with a mapping function. In this work, the Ionospheric Piercing Point (IPP) is set at an altitude of 400 km. Some gaps may be present in TEC data due to ionospheric scintillations. The TEC from Beidou geostationary orbit (GEO) satellites provide pure temporal variations because the IPP of a GEO satellite is motionless. Red triangles in Figure 1 mark the IPPs of BeiDou GEO satellites pair with GNSS receivers (hereinafter referred to as GEO IPPs) selected for analysis in this study. These selected GEO IPPs are located in line along 110°E and 21°N, respectively. This distribution can provide an opportunity to observe detailed latitudinal and longitudinal variations.

Two criteria are being used to measure the amplitudes of INE events in previous studies, as outlined in [46]. At a station or location, we take the minimum value of either TEC or foF2 in the time interval from local sunset to the INE peak as the reference value. ΔTEC or ΔfoF2 is determined from the subtraction of TEC or foF2 from its INE peak value. The INE amplitude—ΔTEC or ΔfoF2—is used to measure the enhancement. The vertical black arrows (Figure 2a,b) mark the ΔTEC and ΔfoF2 during the two events, respectively.

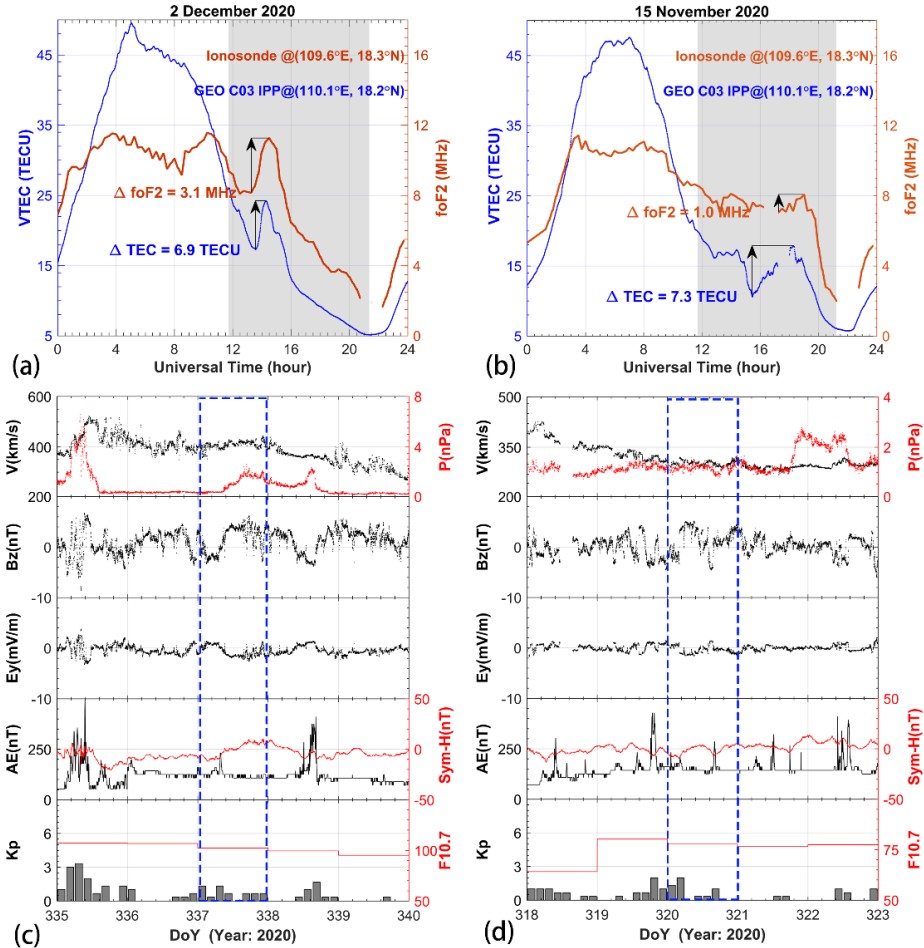

**Figure 2.** Temporal variations of GEO TEC and foF2 values on (**a**) 2 December 2020 and (**b**) 15 November 2020. The blue line plots the values of GEO TEC and orange line gives the values of foF2. The gray-shaded area illustrates local nighttime. The solar and geomagnetic activity indices within 5 days around (**c**) 2 December (DoY 337) 2020 and (**d**) 15 November (DoY 320) 2020, respectively. Shown from top to bottom of panels (**c**,**d**) are solar wind velocity and dynamic pressure, interplanetary magnetic field Bz component, interplanetary electric field Ey component, AE and SYM-H indices, and Kp and F10.7 indices. Blue boxes in panels (**c**,**d**) frame out the time range of 2 December (DoY 337) 2020 and 15 November (DoY 320) 2020, respectively.

To minimize the influence of solar activity, we choose the enhancement events under geomagnetically quiet conditions (see Figure 2). Figure 2c,d shows that the solar wind velocity flies at about 400 km/s on 2 December 2020 and 300 km/s on 15 November 2020, respectively. On the two days, the dynamic pressure of the solar wind is 2 nPa and 1 nPa, and F10.7 index is approximately 102 and 77, respectively. The z component of Interplanetary Magnetic Field (IMF-Bz) is almost northward, and interplanetary electric field Ey is stable. In the meantime, the AE index has the maximum of only 232 nT and 285 nT, indicating that the coupling of the high latitudes ionosphere/thermosphere with solar wind is weak. Kp index is less than 2, and the SYM-H index has the minimum of $-12$ nT and $-10$ nT on 2 December 2020 and 15 November 2020, respectively. Therefore, within the five days around the two events, the solar activity level is low, and the geomagnetic activity is quiet.

## 3. Results

### 3.1. Case A: 2 December 2020

Figure 2a gives the daily variations of GEO TEC, and foF2 observed at Sanya (110°E, 18°N) on 2 December 2020. The blue line represents the values of GEO TEC, and the orange is for foF2. The gray-shaded area illustrates local nighttime. There is a significant enhancement in TEC and foF2 after sunset during 13–15 UT (20–22 LT). The $\Delta$TEC reaches 6.9 TECU at 14:15 UT, and $\Delta$foF2 is 3.1 MHz almost at the same time. They increase about 36% and 40% compared with their reference values. The relative amplitude of TEC observations is slightly smaller than that of foF2 observations.

The temporal variations of foF2 and hmF2 at Mohe, Beijing, Wuhan, Guilin, and Sanya within the time interval from16 LT to 6 LT on 2–3 December 2020 are depicted in Figure 3a. The blue dots show foF2 variations with local time and the orange squares mark hmF2 variations. In Figure 3a, the gray-shaded area indicates local nighttime. The INE event on 2 December 2020 shows a noticeable latitudinal variation. This INE event can be observed at Beijing, Wuhan, Guilin, and Sanya, the black box in panels frames out the evolution of electron density enhancement. At 18:30 LT, the value of foF2 enhances at Beijing station initially, and the hmF2 also rises in this phase. Then, the foF2 enhancement develops toward lower latitudes. The amplitude of INE becomes larger with lower latitude progressively, and both the commence and peak times occur later. During the whole course, hmF2 tends to move to lower altitudes. Two hours later, foF2 exhibits a significant peak at Sanya station. As displayed in Figure 3b, GEO TEC generally shares the change in foF2 on 2 December 2020. This INE event presents a noticeable latitudinal variation along 110°E. The TEC has a larger amplitude of enhancement at lower latitudes and reaches a peak around the IPPs latitude of the receiver HNCM (18.2°N). As marked by black-dashed lines in Figure 3b, there is a considerable time delay with the latitude in both the commencement and peak of the INE. $\Delta$TEC at the northmost station WHHP (28.0°N) rises at 19:00 LT. Two and a half hours later, there are positive values of $\Delta$TEC at HNCM. The ionosphere increases simultaneously at the southern stations. These features are in accordance with the observations from ionosondes. There still are some differences between foF2 and TEC. The enhancement of foF2 can develop at latitudes up to 40°N, while the INE in TEC terminates at the latitude of 28°N. Moreover, the duration of enhancement in TEC is slightly shorter than that of foF2.

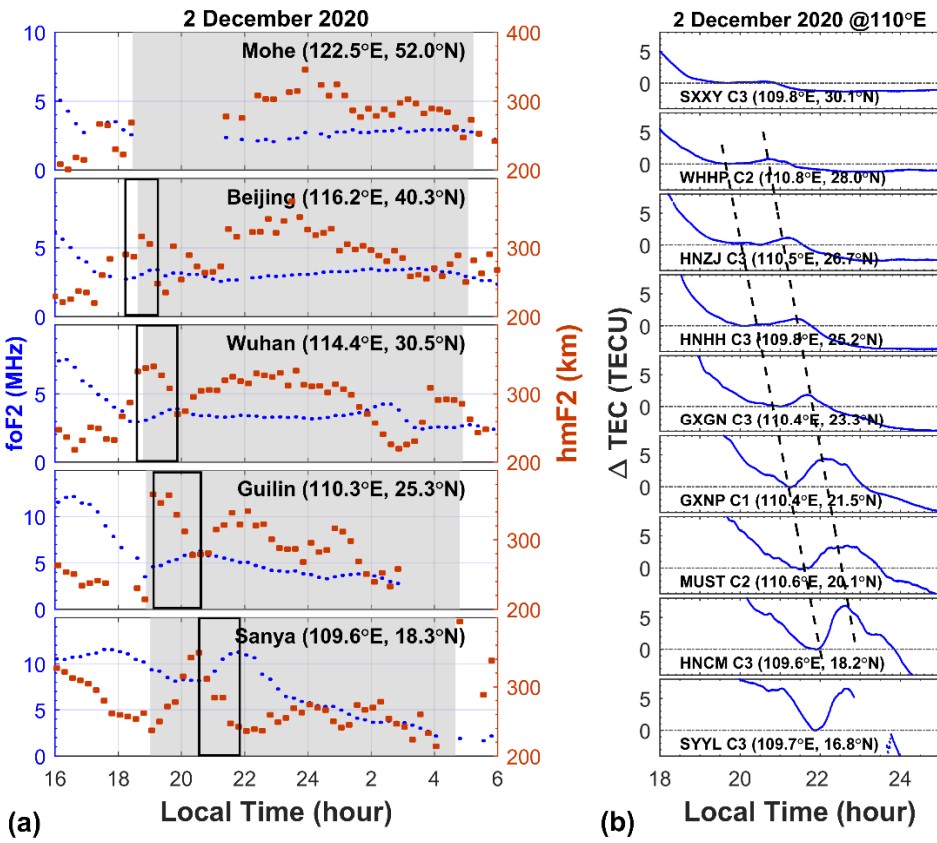

**Figure 3.** (**a**) Temporal variations of foF2 and hmF2 at Mohe, Beijing, Wuhan, Guilin, and Sanya within the time interval from 16 LT to 6 LT on 2–3 December 2020. The blue dots plot foF2, and the orange squares mark hmF2. In panels the gray-shaded area indicates local nighttime and black box frames out the developing phase of electron density enhancement. (**b**) Temporal variations of ΔTEC from 17 LT to 24 LT on 2 December 2020 at different latitudes along 110°E. Black dashed lines connect starting points and peak points separately.

To reveal the spatial features and time evolution of the INE, we select GEO TEC observations from IPPs along 110°E and 21°N during 19–24 LT on 2 December 2020 to produce contour maps, respectively, as shown in Figure 4. By subtracting the reference value of each IPP, contour maps of ΔTEC are also produced. The black dots on the y-axis indicate the locations of selected IPPs. The black line denotes the geomagnetic equator. As shown in Figure 4a, the INE in GEO TEC covers a latitudinal range of approximately 28°N–6°S along 110°E. TEC increases after 19:00 LT around 28°N and develops to 18°N at 21:00 LT. After 21:00 LT, the TEC enhances nearly at the same time at latitudes between 18°N and 6°S. Thus, the maximum latitudinal extent at one epoch is ~24°. Notably, the enhancement mainly occurs in the Northern Hemisphere, although the daytime TEC is higher in the Southern Hemisphere. Moreover, there are longitudinal variations of GEO TEC and ΔTEC. Figure 4b depicts that GEO TEC at 21°N enhances remarkably in a very narrow longitudinal range of 104°E–115°E. The enhancements last at the interval of 20:30–23:00 LT. An interesting feature is that the TEC enhancement occurs earlier on the western side. The largest amplitudes appear at around 18°N and 110°E in latitude and longitude, respectively. As a result, we estimate that the extent of this INE reaches approximately 11° × 34° (longitude × latitude). As regards one epoch, the maximum extent is about 11° × 24° (longitude × latitude).

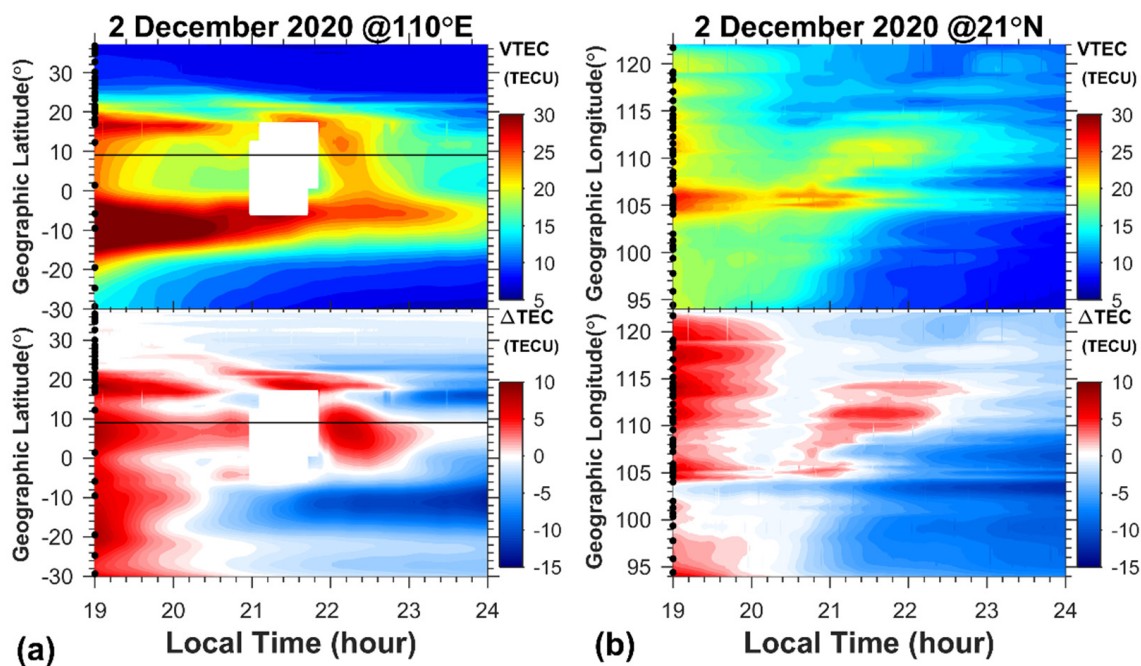

**Figure 4.** Contour maps of GEO TEC and ΔTEC at (**a**) 110°E and (**b**) 21°N during 19–24 LT on 2 December 2020.

*3.2. Case B: 15 November 2020*

GEO TEC and foF2 observed at Sanya on 15 November 2020 are shown in Figure 2b. A postmidnight enhancement in TEC occurred during 15:30–20:00 UT with ΔTEC at about 7.3 TECU, about 63% of TEC reference at 18:17 UT. During the deficiency of foF2, an enhancement is observed at 17:17–20:00 UT with a peak of 1.0 MHz at 19:00 UT.

Figure 5a illustrates that the enhancements in foF2 only occur at Wuhan, Guilin, and Sanya. At 00:20 LT, foF2 at Guilin and Sanya rises almost simultaneously, then foF2 at Wuhan increases at 01:40 LT. The foF2 enhancements at the three stations have similar durations and the amplitude of enhancement at Guilin is the largest. In the course of the enhancements, hmF2 always drops. Temporal variations of ΔTEC at different latitudes along 110°E are revealed in Figure 5b. They show some different features from that of foF2. The enhancement occurred in the southern side of 28°N, with the largest amplitude at IPP latitude of the receiver MUST (20.1°N). Start times and peak times of this INE are identical at different latitudes, except ΔTEC at IPP of the HNMJ (9.0°N) becomes positive slightly earlier than at other latitudes.

From Figure 6, we observe clear spatial variations at 110°E and 21°N. Along 110°E, this INE covers 3°N–28°N with its maximum amplitude around 20°N. It occurs at different latitudes simultaneously and has similar durations. Similar to Case A, the INE in TEC concentrates on the Northern Hemisphere even though the TEC is higher on the Southern Hemisphere. In addition, the longitudinal range of this INE event along 21°N is 103°E–120°E with a maximum amplitude around 110°E. More distinctive longitudinal variation on the local time is present in this INE than Case A. TEC increases around 103°E, then develops to the eastern side about 120°E with a 3-hour time delay. The durations of enhancements at different longitudes are approximately the same. In common with Case A, this INE appears in a very narrow longitude interval, and reaches maximum at around 20°N and 110°E in latitude and longitude. This INE is estimated to cover 17° in longitude and 25° in latitude. For a specific epoch, it covers about 10° in longitude and 25° in latitude.

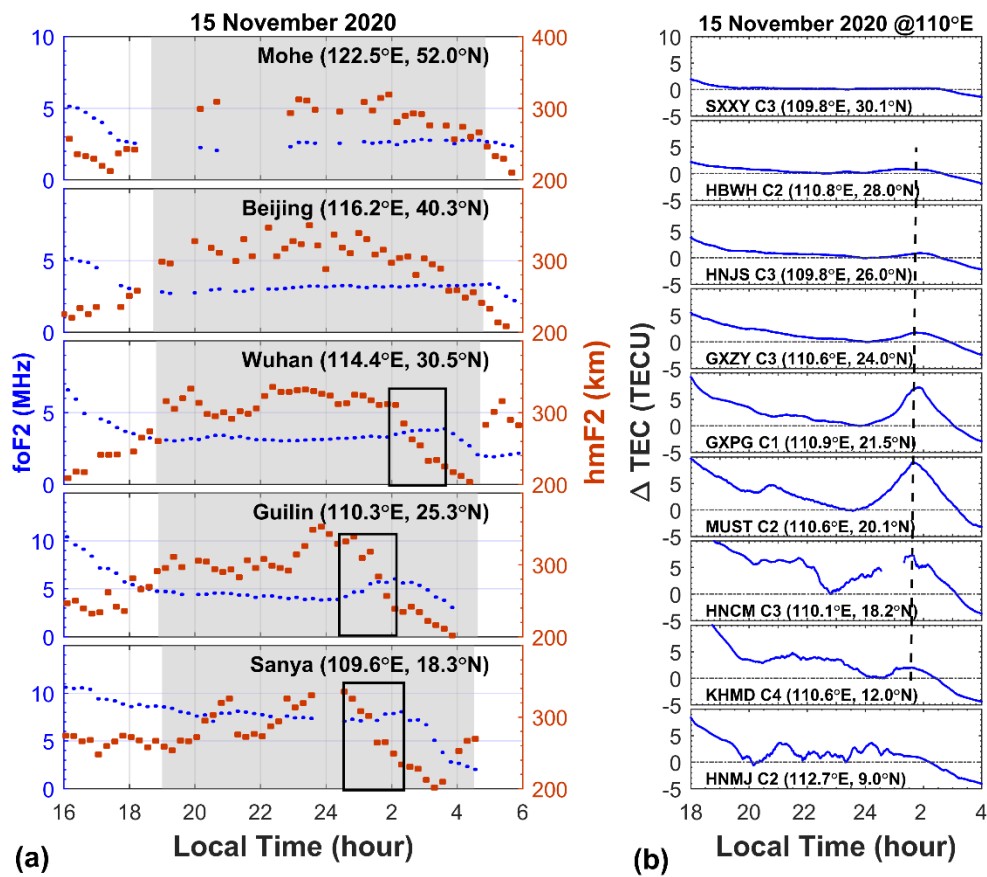

**Figure 5.** (**a**) Temporal variations of foF2 and hmF2 at Mohe, Beijing, Wuhan, Guilin, and Sanya from 16 LT on 15 November 2020 to 6 LT on16 November 2020. (**b**) Temporal variations of ΔTEC at different latitudes along 110°E from 18 LT on 15 November 2020 to 04 LT on 16 November 2020.

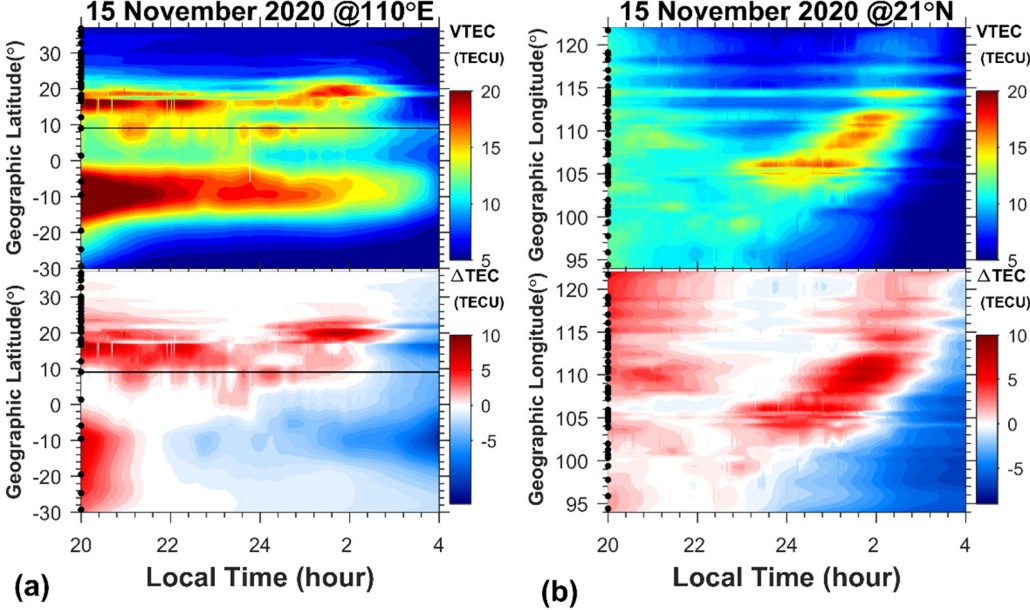

**Figure 6.** Contour maps of GEO TEC and ΔTEC (**a**) at 110°E and (**b**) at 21°N from 20 LT on 15 November 2020 to 04 LT on 16 November 2020.

### 3.3. Two Cases in GIM

Contour maps of TEC from seven GIMs during two cases are presented in Figure 7, as latitudinal variations of GIMs TEC along 110°E and the longitudinal variations along 20°N are exhibited in Figures 4 and 6. Compared with observed GEO TEC, the TEC from the seven GIM models mainly reproduces the basic morphology of the ionosphere, even though there are few differences among them. From Figure 4a, Case A shows the evolution of latitudinal structure of the ionosphere. It presents an equatorial ionospheric anomaly (EIA) signature. The southern crest of the EIA is stronger and lasts a longer duration than the northern crest. The TEC form CODG, JPLG, UPCG, ESAG, and WHUG maps almost replicate these features (see Figure 7a). Case B also experiences an EIA signature. As shown in Figure 6a, the southern crest of the EIA is stronger and almost exists until 04:00 LT. In this case, only CODG and JPLG maps perform well in Figure 7b. In general, CODG and JPLG maps perform best in reproducing the latitudinal structure of the ionosphere (EIA) during these periods. In addition, the longitudinal variations of GIMs during two cases also resemble the variations of observed TEC, besides the longitudinal gradients of CARG maps are more complex than others during the nighttime of 15 November 2020.

However, the GIM models from seven IAACs exhibit unfavorable performance in the two INE cases. Only three GIMs show some weak signatures of INEs. As shown in Figure 7b, the TEC from JPLG maps represents an undeveloped enhancement in a short time at around 110°E on 15 November 2020. A more prominent enhancement occurs in EMRG maps on 15 November 2020, which covers 90°E–105°E with two-and-a-half hours. As shown in the CARG maps from Figure 7a,b, the enhancements occur around the geographic equator at 22:00 LT on 2 December 2020 and at 21:30 LT on 15 November 2020, respectively. The enhancement time on 15 November 2020 is inconsistent with that of the observed INE event; therefore, only the CARG on 2 December 2020 registered the INE. However, the morphology of INEs in the three GIMs is far from the observed TEC. In the two cases, other GIM TEC maps do not display any significant characters about the INEs. This result illustrates that the GIM models are not accurate enough to reliably represent the INEs at low latitudes, which are detected precisely through ground-based observations.

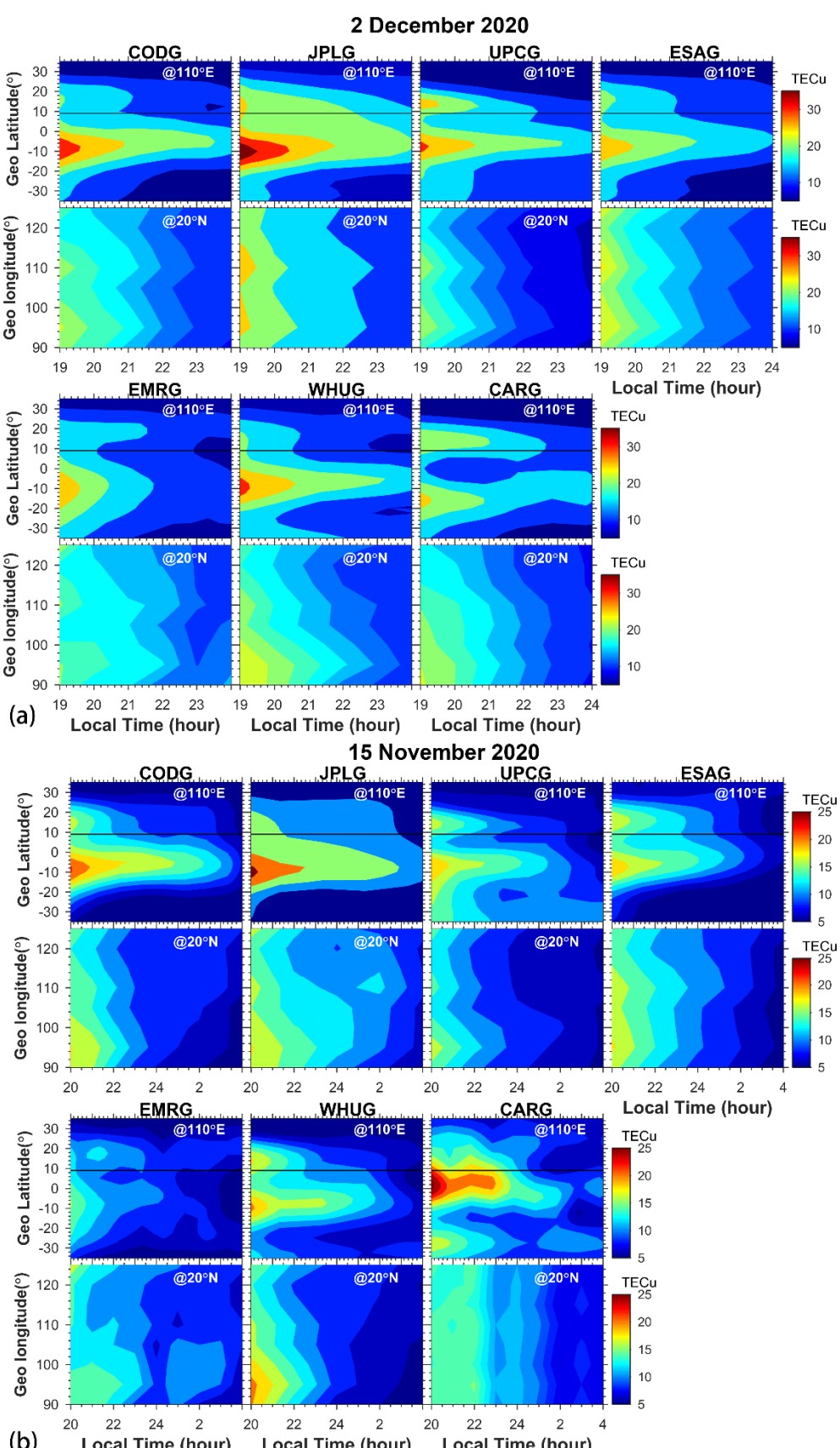

**Figure 7.** Contour maps of TEC from CODE, JPL, UPC, ESA, EMR, WHU, and CAS GIMs at 110°E and 20°N (**a**) during 19–24 LT on 2 December 2020 and (**b**) from 20 LT on 15 November 2020 to 04 LT on 16 November 2020.

## 4. Discussion

### 4.1. Performance of GIMs

The spatial representations of GIM models largely depend on their generation techniques. Especially in the area with sparse GNSS receivers, the spatial resolution of methodology determines the performance of GIMs. As mentioned in Section 2, the techniques proposed from individual IAACs are different. For CODE, ESA, and EMR, the global TEC distribution is described by an SH expansion up to degree and order 15 [3]. To improve the accuracy of GIMs, CAS and WHU combine the SH function with generalized Trigonometric Series functions and inequality-constrained least squares individually. The maximum degree (*N*) and maximum order (*M*) dictate the spatial resolution of SH function:

$$\Delta\beta = \frac{2\pi}{N} \tag{1}$$

$$\Delta s = \frac{2\pi}{M} \tag{2}$$

where $\Delta\beta$ is the resolution in latitude and $\Delta s$ is the resolution in longitude. As the maximum degree and order are 15, the spatial resolution of these GIM models is 24° in latitude and longitude in the area, with few or no GNSS observations. As estimated in Sections 3.1 and 3.2, the maximum extent of INE at a specific epoch is about 11° × 24° (longitude × latitude) in Case A and 10° × 25° in Case B, respectively. The spatial size of two INEs does not meet the spatial resolution of a degree and order 15 SH function. Based on the spatial size of two events in this study, the maximum degree and order of SH function should upgrade to 36 to identify INEs. Moreover, EMR generates global maps using GNSS measurements from roughly 350 stations, despite other IAACs employing 200–300 GNSS receivers. This could be the reason why the TEC from EMRG maps exhibit characters of the INE in Case B.

As for UPC, their tomographic model decomposes the ionosphere into 10° × 5° (local time and latitude) cells and assumes that the electron density is constant in the cells [2]. After interpolating the final grid points, the result is smoothed with a spatial domain of 10° × 2.5° in local time and latitude [38]. With these steps, the fine structures of ionosphere are filtered out. Thus, the spatial sizes of model cells and smoothing window adopted by UPC should be diminished. JPL forms the global maps by interpolating TEC within individual equilateral triangular tiles. The side length of each tile is ~800 km. The intra-tile TEC can be expressed as a linear combination of the TEC at each vertex of the triangular tile [1]. This methodology has a higher spatial resolution so that the TEC from JPLG maps show some signatures about INE on 15 November 2020.

The INE events develop within a short duration; therefore, the temporal resolution of GIMs also affects their performance on INEs. The temporal resolution of different GIM models ranges from 2 h (JPLG, UPCG, ESAG, WHUG), 1 h (CODG, EMRG), and 30 min (CARG). The observed GEO TEC data shows that the INE persists around 2 h in Case A and approximately four and a half hours in Case B. The duration of the INE event in Case A is so short that only CARG maps show some signs of INE on 2 December 2020. The INE event in Case B has a longer duration, which is longer than all of time intervals of different GIMs. Combined with the spatial resolution of each GIM, the JPLG maps and EMRG maps represent part of characters of the INE on 15 November 2020. Accordingly, a high time resolution (time interval shorter than 30 min) is recommended for the applications of GIMs.

### 4.2. Horizontal and Vertical Variations of INEs

The two INE events in this study exhibit dramatic latitudinal and longitudinal features in both TEC and NmF2. Both events appear in a specific region of the ionosphere: the EIA region. The development of EIA makes this region complicated and variable. The INEs at low latitudes are often considered to be related to the EIA evolution. To clarify the relevance between the studied INE events and EIA dynamics, a series of cross sections of EIA are made from TEC observations. The INE event in Case A exhibits notable latitudinal variations of local time of enhancement in both foF2 and GEO TEC. The commencement

and peak of INE occur later at lower latitudes. However, as shown in Figure 8a, the location of EIA crest barely moves, and the northern peak of EIA slowly weaken. It rules out that this INE event arises from the movement of EIA crest. Unlike the Case A, the northern crest of EIA strengthens significantly during 01:30 LT–02:30 LT in Case B. The latitudinal range and time of this crest strengthening are consistent with the INE event in GEO TEC. Obviously, the INE event in Case B is related with the EIA dynamics.

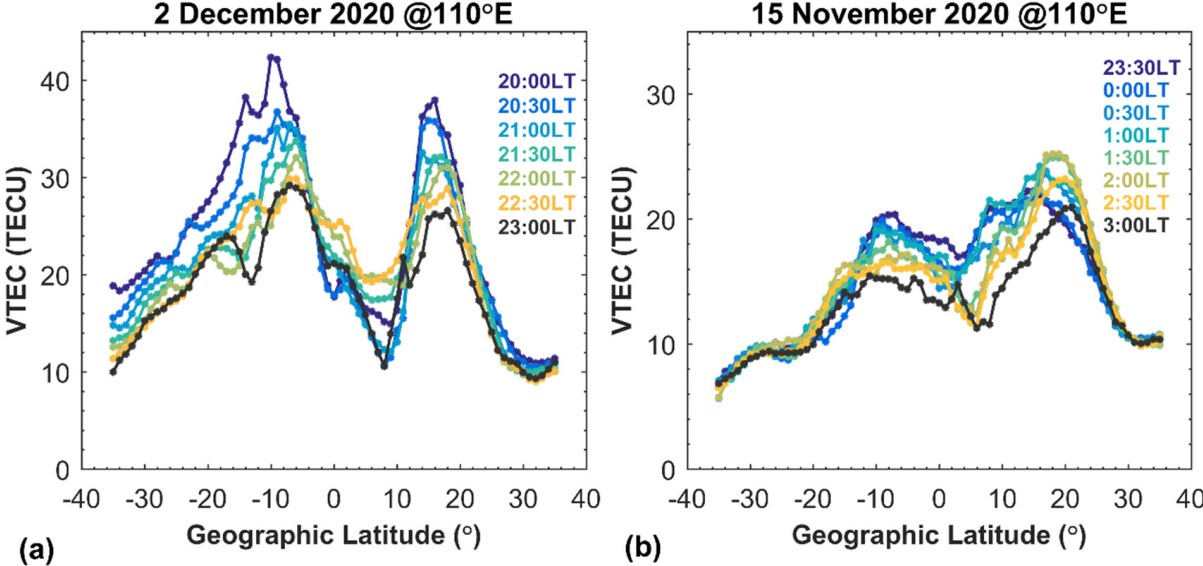

**Figure 8.** Temporal evolution of cross sections of latitudinal structure of TEC observations along 110°E during the INE event on (**a**) 2 December 2020 and (**b**) 15 November 2020.

The two INE events are both limited in a very narrow longitudinal interval: 11° and 17°, respectively. This result differs from the findings in [47]; they reported that the spatial size of INE in GPS TEC is less than 5° in longitudinal. This discrepancy mainly results from the used TEC data. The GPS TEC is unable to distinguish the temporal and spatial variation of ionosphere, whereas our results give more accurate spatial features from GEO TEC. The INEs occur in so narrow regions means the localization of the source. It is difficult to interpret this localization at the current stage.

It is stimulating that the INEs also experience vertical variations. From Figure 9, the electron density height profiles at Sanya in two INE events reveal outstanding altitudinal features. During the enhancement phase, the hmF2 moves down and the topside electron density keeps decreasing. The electron density increases at all altitudes below F2 peak in Case B but only rises around the peak region in Case A. As noted by previous studies, the INEs at low latitudes always present considerable altitudinal variations. The premidnight enhancements at low latitudes are caused by upward movement of the ionosphere in the former hypothesis. The pre-reversal enhancement of the eastward electric fields raises the F layer plasma to higher altitudes with less recombination loss and gives rise to the enhancements of plasma density. The downward shift in Case A is contradictory with this hypothesis. Evident downward movements are observed in postmidnight enhancements at low latitudes by different authors. Liu et al. [46] depicted the altitudinal variations in postmidnight enhancements at low latitudes. Accompanying the development of enhancement in NmF2, the hmF2 descends distinctly, and the height profiles of electron density become thinner. With the descending hmF2, the electron density maintains decaying in the topside ionosphere and increasing at the bottom. These features are fully in accord with the characteristics in Figure 9. Thus, the downward plasma could be the source of the enhancement.

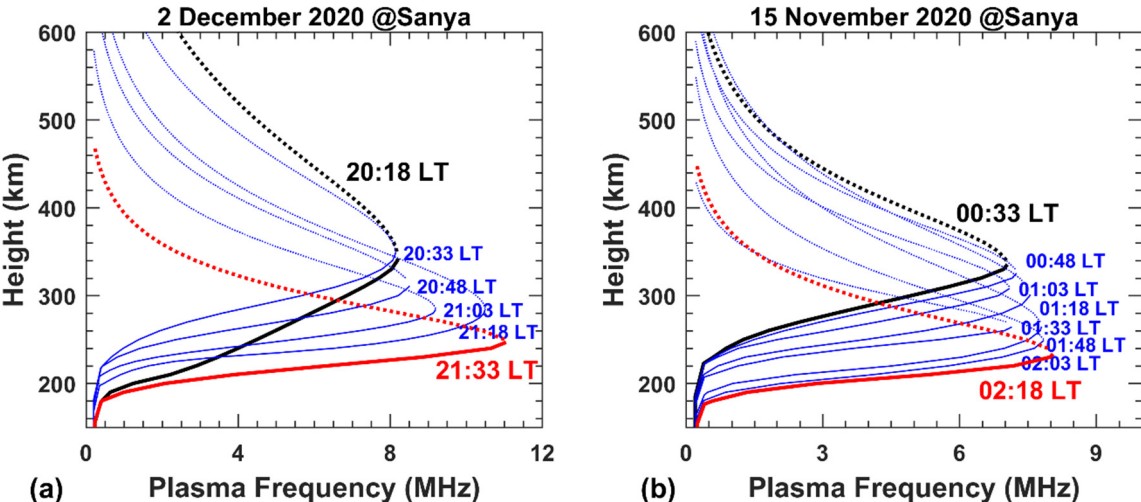

**Figure 9.** Temporal evolution of electron density profiles derived from ionograms recorded at Sanya (109.6°E, 18.3°N) during the electron density enhancement phase on (**a**) 2 December 2020 and (**b**) 15 November 2020. The dotted lines represent topside profiles.

## 5. Conclusions

We present two INE events at low latitudes in the East Asian-Australian sector under geomagnetically quiet conditions, the premidnight event on 2 December 2020 (Case A) and the postmidnight event on 15 November 2020 (Case B), respectively. We utilize five ionosondes data and TEC measurements from Beidou GEO satellites and reveal the detailed spatial pictures of INEs for the first time. By means of the two INE events, the performances of seven GIMs from the ionospheric viewpoint are assessed by comparing with GNSS TEC measurements. In general, the seven GIM models all reproduce the major structures of the ionosphere. However, these models are unable to represent the more rapidly varying ionosphere, such as INEs.

The two INE events display numerous characters from ionosondes and GEO TEC measurements. The premidnight INE in Case A ranges from 28°N to 6°S in latitude and from 104°E to 115°E in longitude with a maximum duration of 2 h. At a certain epoch, this INE has the maximum extent of about 11° in longitude and 24° in latitude. The postmidnight INE in Case B has the latitudinal range of 3°N–28°N and longitudinal range of 103°E–120°E and persists around four-and-a-half hours. This INE has an extent with about 10° in longitude and 25° in latitude at a given epoch. The two events reach the largest amplitude both at about 110°E in longitude and 20°N in latitude. In the two events, the INEs always occur earlier in the western side than the eastern side. The enhancement commences and peaks later at lower latitudes in Case A and is consistent at different latitudes in Case B.

TEC from seven GIMs has similar major structures with the observed TEC as in Case A and B. Among them, CODG and JPLG maps show better performances in representing the latitudinal structure of the ionosphere (equatorial anomaly). However, there is no signature to identify INEs in these GIMs. The extents and durations of the two INEs miss the spatial and temporal resolutions of these GIM models. The generation techniques of these GIM models are the key to their spatial representations. To reproduce the INE events, the SH function, which is adopted by most models, needs to upgrade the maximum degree and order to 36 to represent the TEC distribution with a spatial resolution about 10°. The 3D pixel-based methods developed by JPL and UPC need to improve their meshing and smoothing. To meet the durations of INEs, the time interval of these GIMs is advised to be shorter than 30 min. We hope that these suggestions could be taken into account to improve the GIMs in the future.

**Author Contributions:** Conceptualization, Y.Y. and L.L.; data curation, X.Z. and H.X.; formal analysis, Y.Y.; methodology, Y.Y.; validation, R.Z. and W.L.; writing—original draft, Y.Y.; writing—review & editing, L.L., Y.C., H.L. and M.A.T. All authors have read and agreed to the published version of the manuscript.

**Funding:** This research was supported by National Natural Science Foundation of China (42030202), National key research and development program (2017YFE0131400), National Natural Science Foundation of China (42174204) and the Open Research Project of Large Research Infrastructures of CAS—"Study on the interaction between low/mid-latitude atmosphere and ionosphere based on the Chinese Meridian Project".

**Data Availability Statement:** The Meridian Project of China provided funding in the ionosonde chain. The ionosonde observations used in this paper can be found at URL: http://www.geophys.ac.cn/ (last accessed 2 February 2022). The observed TEC data can be archived from World Data Center for Geophysics, Beijing (http://www.geophys.ac.cn/, last accessed 2 February 2022) and IGS FTP site (ftp://gdc.cddis.eosdis.nasa.gov/, last accessed 2 February 2022). The GIM TEC data were also accessed from IGS FTP site. The 1-min averaged SYM-H, V, P, Bz, and Ey data are downloaded from the OMNI database available at https://omniweb.gsfc.nasa.gov/ (last accessed 2 February 2022). The Dst and F10.7 index are downloaded from the National Geophysical Data Center website (https://www.ngdc.noaa.gov, last accessed 2 February 2022). AE and Kp indices are archived at World Data Center for Geomagnetism, Kyoto (http://wdc.kugi.kyoto-u.ac.jp/index.html, last accessed 2 February 2022).

**Acknowledgments:** The authors acknowledge the use of data from the Chinese Meridian Project and appreciate Wenjie Sun and Lianhuan Hu for providing the GNSS data.

**Conflicts of Interest:** The authors declare no conflict of interest.

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
