# Peer review of "Ionospheric Nighttime Enhancements at Low Latitudes Challenge Performance of the Global Ionospheric Maps"

_remotesensing, doi:10.3390/rs14051088_

Round 1

Reviewer 1 Report

The paper is quite well designed, the material is described consistently, the results are presented clearly. However, it is necessary to adjust the manuscript according to the comments made and to answer the questions raised. The authors are not obliged to discuss the mechanism of the phenomenon under study. However, they should point readers to the problems associated with the specificity of the studied area of the ionosphere. In any means.

Reviewer 2 Report

The paper presents a case study analysis of two INE (ionospheric nighttime enhancement) events observed in autumn/winter of 2020 in Asian region. I event is the pre-midnight INE and another is the post-midnight INE. The ionosphere is studied using TEC, f0F2, hmF2 and NmF2 parameters obtained from ionosondes and GNSS receivers. Temporal and spatial variations of the studied parameters are analysed.

Even though this is a case study, it shows interesting features of the night-side ionosphere. 

Comparison of the observational data with 7 widely used GIMd (global ionospheric maps) allowed authors to define the spatial and temporal resolution of GIMs that may allow correct reconstruction of INEs.

 Interesting observations on the changes of the electron density in the ionosphere  associated with INEs are also presented. Since some of the observed features of the electron density height profiles contradict with previous studies, it would be good to check them, e.g., with observations of the top-side ionosphere from satellites for the studied events. Such comparison  can be used as a support of the hypothesis presented in ll. 353-358.

I recommend accepting the paper after major revision since not just English needs to be thoroughly checked but also formatting of the text (some subscripts are missing) and figures. For example:

Ll. 86-87:  Please rephrase: like this it hints that you analysed many events and can make reliable conclusions on the average the behaviour of INEs, while in reality you present cases studies

Fig. 2: please make correspondence between the time axes in a (b ) and c (d), e.g. by connecting edges of the X-axes of a  &  b to corresponding time intervals in c & d. Do you really need to show solar wind & geomagnetic parameters for 2 days after the studied day?

  1. 237: “They show some different features.” Different from what?

Ll. 249-250: “Some distinctive longitudinal variation on the local time of this INE than Case A.” Please rephrase

Fig. 7: I believe it would be better to organise GIMs not by their sources but by events?

Fig. 8: If the time variations (profiles at different times) were shown by different colours or line types (with a legend attached or in captions), the time evolution of the height profiles of the electron density will be clearer.

Reviewer 3 Report

Referee’s report on

Ionospheric Nighttime Enhancements at Low Latitudes Challenge the Global Ionospheric Maps

by Yuyan Yang et al.

This is a study of two events involving nighttime enhancements of ionospheric plasma density (INE) that occur on scales (spatial and temporal) that are finer than the standard resolution of the common Global Ionospheric Map (GIM) provided by the institutions of the International GNSS Service (IGS) can describe. The authors argue that, with the increased reliance on products such as the GIM, it is time to improve the standard resolution of the GIM to be able to describe events such as these. There are a number of ways in which the presentation should be improved:

The method by which the ‘contour’ maps of TEC in Figure 4 were produced is not described. It presumably is a technique similar to those used in making the IGS GIM maps, because the same kind of data serves as the input and a similar output is created. This should be presented. What are the white rectangles in the plot? They fall in an important region of the plot as far as the INE are concerned, so if they’re due to missing data or some problem with the analysis, perhaps a different event without the problem there should be considered. The same color bar range should be used for the Lat-Time and Lon-Time plots so that it is easier to match features.

Figure 7 needs reorganizing, putting the maps for each event in separate Figures, with the Lat-Time and Lon-Time maps from each institution side by side like in Figure 4. Again, the same color-bar ranges should be used for all the plots for each event.

As the authors note, the topside profiles in Figure 8 are fictitious; ionosondes can give no information on the structure of the density profile above the peak. I believe that the providers of such profiles simply choose the height scale of the topside to match the height-scale of the bottomside, so when the bottomside becomes compressed, so does the topside in their portrayal. Given that, the authors should not belabor the topside structure shown here; in fact, if the topside portion of the profile is shown at all, I’d suggest plotting it with a dashed or dotted line, and little discussion of the topside is  warranted.

Finally, when you speak of increasing the time resolution, there is an ambiguity: does that mean making the time intervals larger? Better to say refine the time resolution or make the time resolution finer.

Round 2

Reviewer 2 Report

hank you very much for the revision. I have no more comments.

Reviewer 3 Report

Publishable as is